# Latent forward model for Real-time Strategy game planning with incomplete information

## Abstract

Model-free deep reinforcement learning approaches have shown superhuman performance in simulated environments (e.g., Atari games, Go, etc). During training, these approaches often implicitly construct a latent space that contains key information for decision making. In this paper, we learn a forward model on this latent space and apply it to model-based planning in a miniature Real-time Strategy game with incomplete information (MiniRTS [Tian et al. (2017)]). We first show that the latent space constructed from existing actor-critic models contains relevant information of the game, and design training procedure to learn forward models. We also show that our learned forward model can predict meaningful future state and is usable for latent space Monte-Carlo Tree Search (MCTS), in terms of win rates against rule-based agents.

## 1 Introduction

Model-free deep reinforcement learning (DRL) approaches (e.g., deep Q-learning [Mnih et al. (2013)], DDPG [Lillicrap et al. (2015)], A3C [Mnih et al. (2016)], etc) have been applied extensively in many simulated environments with complete information and relatively simple game dynamics (e.g., Atari games, Go [Silver et al. (2016; 2017)], Doom, etc). The learned agent, which acts reactively based on the current game situation, can even achieve superhuman performance.

However, for complicated environments, planning ahead (or "predicting the future") before making an actual decision is important. Such a planning procedure requires a *forward model* that estimates the next state $s_{t+1}$ given the current state $s_t$ and action $a_t$, which is in general non-trivial to construct and estimate from the high-dimensional raw input. For partially observable environments (e.g., Real-time Strategy Games like StarCraft), constructing a forward model is more difficult even with a perfect domain knowledge of the game, due to the deliberate concealing of information and the additional requirement to capture the belief of the unknown for the agent.

A natural question now arises. Could we borrow the success of model-free approach to learn a forward model? Note that in model-free approaches, a single shared network (called "trunk") is often used to extract features from the input game situation to obtain a latent representation. From the latent space, multiple reinforcement learning quantities ($Q$-function, value function $V$, advantage function $A$, etc) are predicted via simple linear transformations and used for decision making. Strong performance of these approaches indicates that the learned latent space must have captured key ingredients of the input situation and remains low-dimensional. Therefore, it is an excellent candidate for the state representation of a forward model.

In this paper, we study whether it is possible to use the latent space learned by model-free approaches to construct forward models. We use MiniRTS [Tian et al. (2017)], an efficient and simple two-player Real-time Strategy (RTS) game. MiniRTS captures the basic dynamics of its kind: the agent builds units (workers and troops) that consume resources, gathers resources, explores regions out of sights ("fog of war"), defends enemy's attack, and invades enemy's base. This game is incomplete information, because the agent can only see within its sight, and does not know the action of its opponent by default. Rather than unit based control as in [Synnaeve et al. (2016); Usunier et al. (2016); Vinyals et al. (2017)], the agent uses 9 discrete actions to control the overall strategy (e.g., build a particular kind of troops, attack or defend).

Our contributions are three-fold: First, we propose to study the relationship between the latent space learned by model-free approaches and the state representation of forward models. Very few works (e.g, DARLA [Higgins et al. (2017)], DQN [Mnih et al. (2015)]) in model-free RL study these properties in depth, let alone using the latent state in model-based approaches for incomplete information game. To our knowledge, we are one of the first works to explore such directions. Second, we improve the performance of model-based agent in MiniRTS by input feature design and show that the latent space learned from actor-critic models [Mnih et al. (2016)] can reconstruct critical information of the game, e.g., Hit Point of the base and available resources. Finally, we propose novel algorithms that learn a forward model that maps a latent state $h_t$ to its future counterpart $h_{t'}$ ($t' > t$) with reduced drifting. Such a forward model enables us to use model-based planning such as Monte-Carlo Tree Search (MCTS) in incomplete information games. We show positive performance (8% higher than random planning) in terms of win rates against rule-based agents.

## 2 RELATED WORK

**Forward modeling** Model-based approaches are one of the standard ways to model complicated yet physically calibrated robots [Grizzle et al. (2010)]. In these cases, it is assumed, or known by design, that a forward model can be obtained easily. Furthermore, the state representation of the forward model is usually the internal state of the system (e.g., position and velocity of each robot joint), and is thus low-dimensional.

On the other hand, learning a forward model from a complicated and high-dimensional observation is in general difficult, even with the current deep learning architecture. Computer vision researchers try to predict the next video frame given the last few frames [Mathieu et al. (2015); van Amersfoort et al. (2017)]. To reduce the complexity, one natural idea is to project the high-dimensional input to low-dimensional state that captures the important aspect of the situation, on which a forward model might be easier to learn. [Agrawal et al. (2016)] learns a forward model directly from visual input and uses that for manipulation. To make their learned latent state nontrivial, a regularization term is applied to ensure the learned representation can reconstruct the input as much as possible. In comparison, we do not need a regularization since the latent space is pre-trained by model-free approaches.

One issue in forward modeling is that its accumulative prediction over many steps might drift and is not usable for planning. [Weber et al. (2017)] uses forward model prediction as another feature for a model-free approach, while Bansal et al. (2017) uses forward model as a way to guide exploration for model-free procedure. In this paper, we use multi-step long-term prediction to stabilize the forward model.

**Real-time strategy games.** Most works on Real-time Strategy (RTS) games assume that the agents have access to complete information, in which the behaviors of both the player and the opponents are observed. Micro-management is often viewed as a standard RL task, addressed by model-free approaches [Usunier et al. (2016); Peng et al. (2017)]. On the other hand, Monte-Carlo Tree Search (MCTS) has been applied (e.g, ABCD [Boutilier et al. (1997)], MCTSCD [Soemers (2014)]), which focus on concurrent planning with perfect information and perfect forward model. [Uriarte & Ontanón (2015)] learns a short-term shallow forward model for local combat, but again assuming full access to the game. Finally, [Butler & Demiris (2010)] deals with partial observability, but focuses particularly on scouting, rather than global strategies as we do.

Grouping unit-based actions into macro (or strategic) actions is an open and challenging topic in RTS games. Following [Tian et al. (2017)], we use 9 pre-defined discrete strategic actions to facilitate training. Other recent works also model macro strategies from replays [Justesen & Risi (2017)] or from action scripts [Barriga et al. (2017)], both with deep models.

## 3 METHODS

In model-free reinforcement learning, an agent learns to maximize its long-term reward, either from previous experience in the same environment (e.g. replay buffer), or from its own interaction with the environment with its behavior. In both cases, the agent does not need to have knowledge of the environment, nor know how the world changes after an action is taken, hence called model-free

approach. On the other hand, if an agent learns to build a model to predict how the state (its own state and/or environment state) changes after its action, then the agent is considered to take a model-based approach.

In this paper, we first use an off-policy version of Batch A3C [Wu & Tian (2017); Babaeizadeh et al. (2017)] to train a model-free agent. During training, the parameters of the agent follows the gradient directions of the two objectives:

$$\Delta\theta_\pi \quad \propto \quad \eta_t\nabla_\theta\log\pi(a_t|s_t;\theta_\pi)(R_t - V(s_t;\theta_V)) \tag{1}$$

$$\Delta\theta_V \quad \propto \quad (V(s_t;\theta_V) - R_t)\nabla_{\theta_V}V(s_t;\theta_V) \tag{2}$$

where $\pi(a|s;\theta_\pi)$ and $V(s;\theta_V)$ are the policy and value functions of the agent. $\eta_t = \frac{\pi(a_t|s_t;\theta_\pi)}{\pi(a_t|s_t;\theta_\pi^{old})}$ is an importance factor that captures the discrepancy between the policy distribution $\pi(a_t|s_t;\theta_\pi^{old})$ evaluated when we sample the action $a_t$ at state $s_t$, and the policy we use now with the current parameter $\theta_\pi$. We apply the importance factor $\eta_t$ since Batch A3C does not start updating model parameters until it collects a full batch. During that period, the data that are first collected is considered to be off-policy.

In many works that use shared feature extraction and multi-head predictions [Wang et al. (2015); Wu & Tian (2017); Silver et al. (2017)], the two set of the parameters $\theta_V$ and $\theta_\pi$ are mostly shared with a common trunk $h$ with parameter $\theta_U$, and the policy/value function can be written as:

$$\pi(a|s;W) \quad = \quad \mathrm{softmax}(Wh(s;\theta_U)) \tag{3}$$

$$V(s;\mathbf{w}) \quad = \quad \mathbf{w}^T h(s;\theta_U) \tag{4}$$

$h$ compactly encodes the information used for both decision (policy $\pi$) and evaluation (value $V$) of the model. Compared to the original high-dimensional input that might contain a lot of irrelevant information, $h$ typically contains a rich but low-dimensional representation of the current situation that is critical for its assigned task. That might include world states directly related to short-term/long-term reward, and belief of the unobserved part of the environment. From a model-based point of view, such a latent space learned from a successful model-free training is a good candidate for the state representation of a forward model.

## 3.1 Build a strong latent space

Following the reasoning, in order to build a strong forward model, a natural approach is to start with a strong pre-trained model-free agent that performs well in the environment. In this paper, we use MiniRTS [Tian et al. (2017)] as the environment, which is a fast incomplete information Real-time Strategy Game with two players. An agent which performs well in MiniRTS must build its own troops, gather resources efficiently, explore unseen territory, defend the attack from the enemy, and find the right timing to attack back. The game ends when one player's base is destroyed by the opponent. MiniRTS runs at 40K FPS on a laptop with quite complicated dynamics (e.g., collision detection, unit path planning, etc) and is suitable for our task.

To strengthen the model-free agent, we improve upon [Tian et al. (2017)] by appending more input feature channels. We define three models *Vanilla*, *BuildHistory* and *PrevSeen* , each with a different set of features (Tbl. 1).

- **Vanilla** Similar to [Tian et al. (2017)], we use basic features that include Unit Type, Hit Point ratio, and available resources to build units.

- **BuildHistory** In an incomplete information game, the decision a player made two minutes ago is not the same as the decision the player is making now, even if the perceived situation is identical. This is because during the period, the opponent (which the player cannot see) might have done a lot of work. To make the agent aware of the situation, we record the build_since tick of each unit, and attach related information in the feature plane.

- **PrevSeen** Ideally the agent should remember all information since the game starts (e.g., the opponent's base location it saw at tick 100, even if the current tick is 5000). However, as shown in our experiment, Recurrent Network (RNN) is not able to carry such a long-term memory. Therefore, we modify the game engine so that the most recent information that the agent used to have access to, is also sent into the input. As shown next, it gives

| Model | Feature Name | Description | #Channel |
|---|---|---|---|
| *Vanilla* | Unit Type | 1-hot vector for 6 unit types at $(x, y)$ | 6 |
| | Affiliation | 1 (our unit), 2 (enemy unit) | 1 |
| | HP Ratio | Hit Point / Max Hit Point | 1 |
| | Base HP Ratio | Hit Point / Max Hit Point for the base | 1 |
| | Resource | Resource binning | 5 |
| *BuildHistory* | History decay | 50 / age | 1 |
| | History bin | 1-hot vector for *age* binning | 7 |
| *PrevSeen* | Opponent unit type | 1-hot vector for 6 unit types at $(x, y)$ | 6 |
| | Opponent history bin | 1-hot vector for *age* binning | 7 |

Table 1: Channel used in the experiments. Note that *age* = current_tick - unit_build_tick. Age is binned into $[200, 500, 1000, 2000, 5000, 10000]$ (7 bins). Resource is binned into $[50, 100, 150, 200]$ (5 bins). All models have the same number of parameters. *BuildHistory* includes all *Vanilla* features, *PrevSeen* includes all *BuildHistory* features. Features that are not included in the model are 0.

> much faster convergence, and encourages fast map exploration and changes in strategies. As a result, the bot learns more aggressive strategies like rush by itself.

In Sec. 4.1, we show that *PrevSeen* has the strongest performance and its latent representation is a good candidate for forward models. To check whether the learned latent space is useful for forward modeling and understand what kind of game statistics it has captured, we further learn an interpretation network to predict the input from the hidden state. The interpretation network is designed to mirror the network that maps input to latent space, only replacing pooling layers with upsampling layers. Again, the latent space of *PrevSeen* can predict highly relevant input features, as shown in Sec. 4.2.

## 3.2 LEARN A FORWARD MODEL IN THE LATENT SPACE

Once we have a rich latent space that contain important information for decision making, we thus train a forward model that carries $h_t$ to the near future, which can be used for prediction and model-based planning.

One naive approach is to learn a forward model $f$ to predict the next state $h_{t+1}$ given $h_t$ and $a_t$. However, a simple prediction might suffer from drifting, in which if we apply $f$ multiple times to predict far future, then the predict error will quickly accumulate and makes the long-term prediction unusable. This leads to detrimental performance for planning methods (e.g. MCTS) which require accurate long-term predictions.

To solve this issue, during training, we not only require $f(h_t, a_t)$ to be close to $h_{t+1}$, but also require $f^{(t')}(h_t, a_t, a_{t+1}, \ldots, a_{t+t'-1})$ to be close to $h_{t+t'}$, where $f^{(n)}$ is to apply the same forward model $n$ times. While a naive implementation requires $O(T^3)$ gradient update, such a training procedure can be implemented efficiently with the current autograd approach with only $O(T^2)$ gradient update. The trained forward model enables us to apply existing planning method to complicated incomplete information games and give a sensible performance boost over baselines.

## 4 EXPERIMENTS

### 4.1 TRAIN A STRONG MODEL-FREE AGENT

We first learn a model-free agent against rule-based AI (`AI_SIMPLE` in MiniRTS). To maximize the performance of trained AI, we use $T = 20$ for all the experiments, as mentioned by [Hessel et al. (2017); Firoiu et al. (2017)] that multi-step training with large $T$ is useful. We also set the frame skip (how frequent the action is made) as 50 for AI, and as 20 for `AI_SIMPLE`. This gives the trained agent a slight disadvantage.

For network structure, we use a convolutional networks with 2 conv + 1 pooling + 2 conv + 1 pooling, each convolutional layer has 64 channels, except for the first and the last layer. The input features

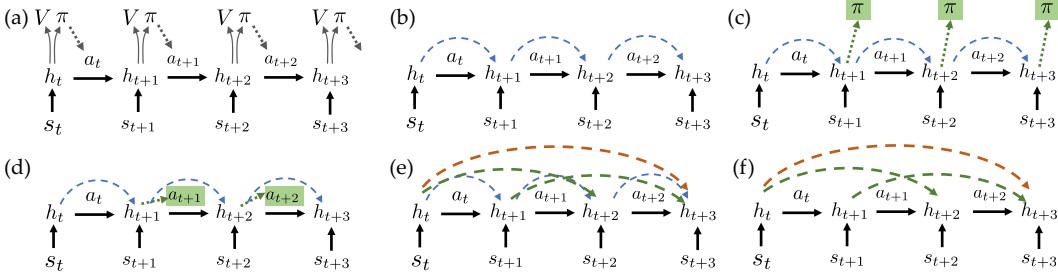

Figure 1: Training forward models on latent space.**(a)** Prediction structure of A3C model. **(b)** Predict the immediate next latent representation $\tilde{h}_{t+1}$ by $h_t$ and $a_t$ and directly minimizes $\|h_{t+1} - \tilde{h}_{t+1}\|$; **(c)** Predict $\tilde{h}_{t+1}$ so that $\tilde{\pi}_{t+1}$ matches with $\tilde{\pi}_{t+1}$, the policy distribution from the true latent state $h_{t+1}$; **(d)** Predict $\tilde{h}_{t+1}$ so that the greedy action from $\tilde{\pi}_{t+1}$ matches with $a_{t+1}$, the action predicted from policy distribution $\pi_{t+1}$; **(e)** Multiple hop predictions. **(f)** Only predict long-hop.

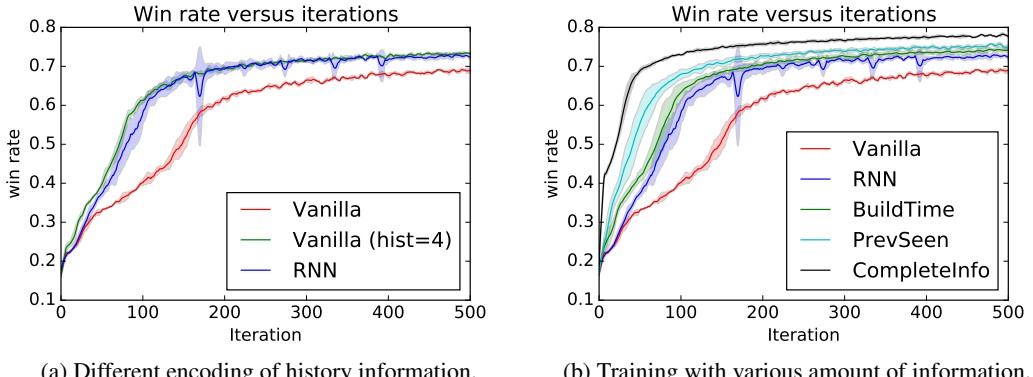

(a) Different encoding of history information.

(b) Training with various amount of information.

Figure 2: Training curves. Each experiment is repeated 3 times. Each iteration contains 250 minibatches with batchsize 128. The win rate is computed by sampling actions from the current policy.

is 20-by-20 images with 35 channels (Tbl. 1). For all approaches, the overall number of parameters is kept constant as in [Weber et al. (2017)]. The resulting latent representation is $35 \times 5 \times 5 = 875$ dimensional, which is much lower than the original input space (14000). After convolution, the latent representation is then used to predict discrete action distribution (policy $\pi$) and value function $V$. For RNN, the next latent state $h_{t+1}$ is computed by first concatenating $h_t$ with a learnable embedding of $a_t$, and then compressing them into a vector in the latent state by MLP. We also use frame-stacking (of size 4), which is an empirical trick to mimic short-term memory during training. Both RNN and frame-stacking use the feature sets of *Vanilla*.

As in [Tian et al. (2017)], we use 9 discrete actions to drive the environments. All these actions are globally strategic actions (e.g., attack, defend, which unit to build) and agents do not need to worry about their details (e.g., where to build the unit).

**History Information.** Adding historic data can help with the performance of RL approach. However, similar to previous works [Wu & Tian (2017)], the performance of frame-stacking is already comparable to RNN approach, showing that the latent space really captures the short-term memory. On the other hand, *PrevSeen* outperforms both RNN and *BuildHistory* . In summary, while the short-term information can be encoded into the hidden space (either by RNN or frame-stacking), long-term information cannot be efficiently learned in the latent space and has to be encoded manually (as in *PrevSeen* ).

**Complete/Incomplete information**. This leads to difference in training curves (Fig. 2) and final performance (Tbl. 2) whether the agent can see the opponent's behavior and whether the agent has immediate access to the information that encodes what it saw long-time before (e.g., sense the base of the opponent long before).

| Method | Vanilla | Vanilla(hist=4) | RNN | BuildHistory | PrevSeen | Complete Info |
|--------|---------|-----------------|-----|--------------|----------|---------------|
| Win rate | 72.9±1.8 | 79.8±0.7 | 79.7±1.3 | 80.8±1.7 | 81.4±0.8 | 81.7±0.7 |

Table 2: Win rate of trained AI with frame skip 50 against the opponent (AI_SIMPLE) with frame skip 20 over 10K games. During evaluation, at each step, the agent greedily takes action of maximal probability.

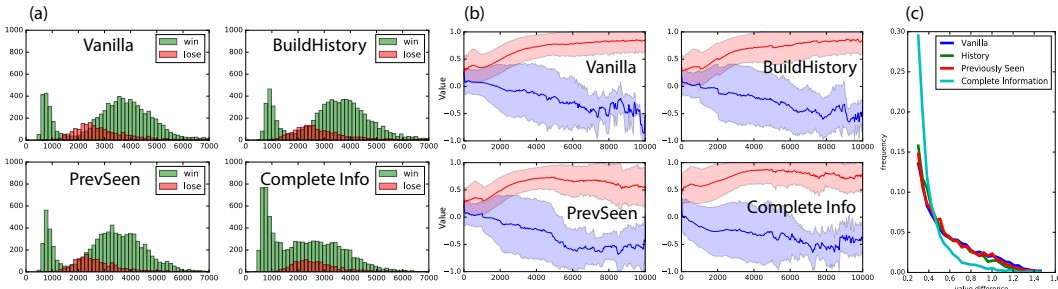

Figure 3: Statistics of 10K games played by AI trained with different input information. **(a)** Game length statistics for won/lost games. **(b)** Value estimation when game progresses. **(c)** Histogram of surprise metric (Eqn. 5). Complete information gives less surprise.

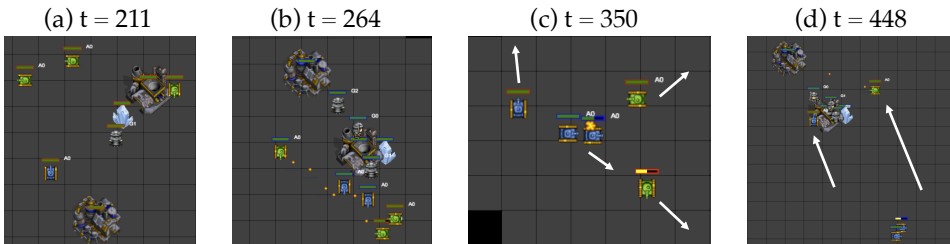

Figure 4: A sample game between *PrevSeen* and rule-based system. **(a)** AI decides to build 2 tanks and rush, without building any workers. **(b)** AI fights with the opponent. **(c)** AI retreats, one tank attracting enemy troops into one direction. **(d)** The other two tanks move back to the enemy base and harass.

**Value function**. In incomplete information setting, we see many sudden drops in the value function when game progresses. This is because the agent does not estimate well about the opponent's actions and thus thinks it is in good shape. To quantify it, we define the surprise metric as the difference between nearby estimated values:

$$Surprise_t = |V(s_t) - V(s_{t-1})| \tag{5}$$

Fig. 3(b) shows that AI trained with complete information indeed knows more about the situations than those trained with incomplete information, in particular at the beginning of the game. As shown in Fig. 3(c), the distribution of surprise is different between complete and incomplete information bot.

However, even for the complete information bot, there are quite a few sudden drops. This is due to the fact of complicated dynamics of the game. Lots of small factors, e.g., path planning, collision detection, could lead to substantial different consequences of the game. For example, the consequence of an even battle is often determined by the micro-management of each player, for which our model does not have control.

**Different learned strategy**. Although the final performance of *PrevSeen* and *BuildHistory* are similar, their behavior against rule-based AI is quite different. As shown in Fig. 4, AI trained from *PrevSeen* learned to explore the map and/or rush the enemy with a few tanks at the beginning of the game. Quantitatively, the number of nonzero seen-then-hidden events per game from *PrevSeen* is twice (3405) than that (1736) from *BuildHistory* (Fig. 5). We haven't seen similar behaviors in prior works.

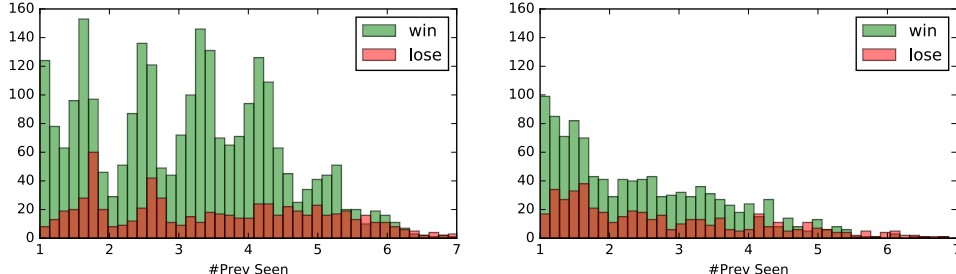

Figure 5: Histogram of averaged number of distinct opponent units seen-and-hidden per decision point from 10K games of *PrevSeen* (left) and *BuildHistory* (right). *PrevSeen* explores more.

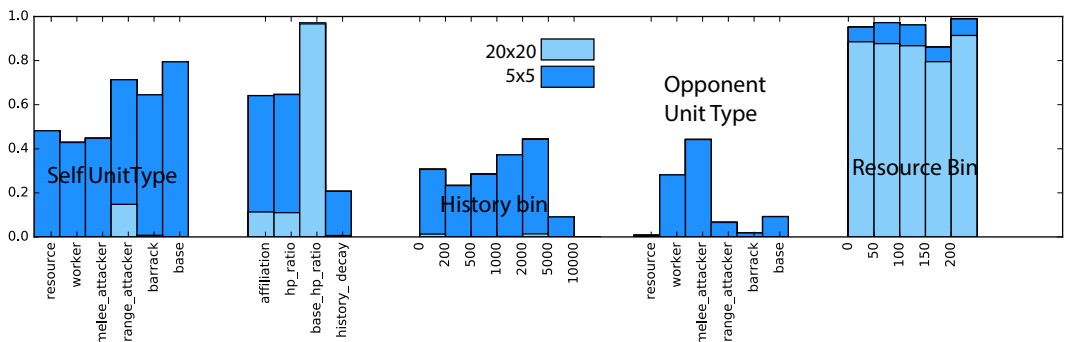

Figure 6: Normalized reconstruction accuracy of each channel.

## 4.2 RECONSTRUCTION FROM THE LATENT STATE

With the latent state, we can reconstruct the original 20x20 channels. We choose to predict one channel at a time. A joint model that predicts all input channels at once works poorly since dense channels (with many non-zero entries) dominate the prediction.

We define the normalized reconstruction accuracy (NRA) to be $1 - \|c - c^*\|/\|c^*\|$, where $c$ is some channel and $c^*$ is its ground truth value. Fig. 6 shows the results. The model correctly finds the most relevant features from the input (e.g., base HP ratio, the amount of available resource). We discover that the agent can partially recover the exact location of the workers during early stage of training, but ignores them at convergence, showing that the model learns to discard less relevant features. Interestingly, the prev-seen feature channels are not predicted well for the full-fledged agent, showing they are not important in the decision. However, models equipped with these channels train much faster.

We also try predicting the 5x5 down-sampled version of the original 20x20 channels, by removing all upsampling layers. Overall the normalized reconstruction accuracy is much higher, as shown in the dark blue bars. This is because the latent state is also a $5 \times 5$ down-sampled state of the original game state, and the model only needs to reconstruct the features in a rough region instead of in each grid. In particular, we see the emergence of multiple relevant features such as the location of own RANGE_ATTACKER and BASE, affiliation and unit HP ratio, etc. We also see that the agent learns to pay attention to the opponent's WORKER (which gathers resources) and MELEE_ATTACKER (which attacks us). The model pays attention to the opponent's MELEE_ATTACKER but not so much to RANGE_ATTACKER because the rule_based AI mostly builds MELEE_ATTACKER.

## 4.3 LEARN A FORWARD MODEL

We train multiple forward models following Sec. 3.2, each with different training paradigm defined as follows. **Pred1**: basic forward model (Fig. 1(b)); **MatchPi**: enforce the predicted future state to predict the future policy well (Fig. 1(c)); **MatchA**: enforce the predicted future state to predict the future action well (Fig. 1(d)); **PredN**: predict long-term future states (Fig. 1(e)).

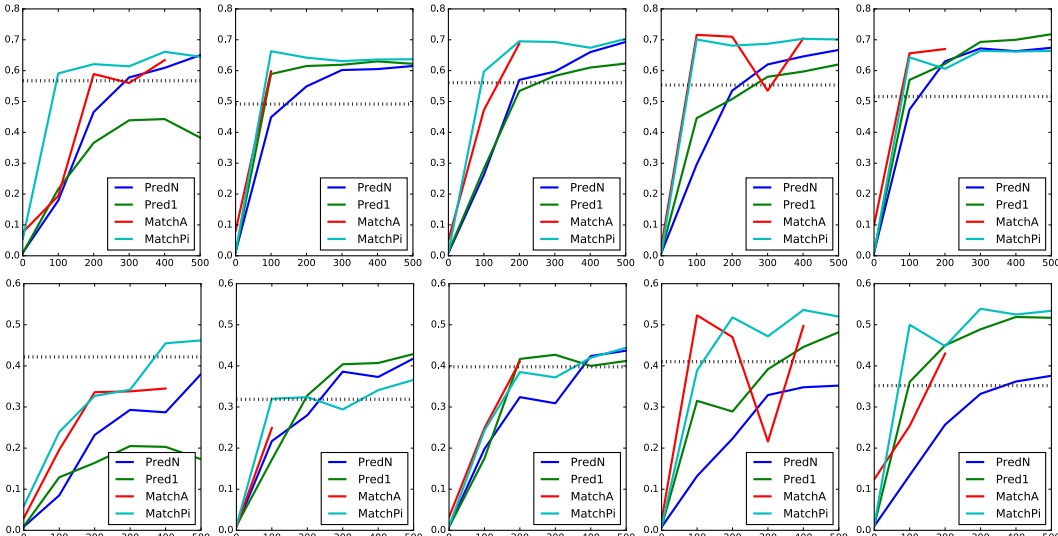

Figure 7: Win rate from the decisions made by latent state $h$ that is predicted by forward models. **Top row:** 2-hop prediction ($\hat{h}_t = f^{(2)}(h_{t-2})$) on 5 models trained on 5 independent A3C models. **Bottom row:** 5-hop prediction. In both cases, the grey dotted line is the baseline $\hat{h}_t = h_{t-2}$ (and $\hat{h}_t = h_{t-5}$). **MatchA**(red) lines do not continue due to numerical instability.

For evaluation, at a time step $t$, we recursively apply the learned forward model to estimate the current latent state $\hat{h}_t$ from previous state $h_{t-t'}$ that is computed from observation $s_{t-t'}$. Then we use $\hat{h}_t$ to make decision for time $t$, and check its win rate against the rule-based system. As a baseline, we use identity forward function $\hat{h}_t = h_{t-t'}$.

We evaluate these training paradigms in 5 independently trained models using model-free approaches. As shown in Fig. 7, **MatchPi** is the most stable method with highest win rate, consistently higher than the baseline approach that uses $\hat{h}_t = h_{t-t'}$. **MatchA** also learns quite fast, but runs into numerical instabilities after a few hundreds of iterations. **PredN** is also quite stable but its learning speed is not as fast compared to **MatchA**. When the delay is large, we clearly see the decay of the win rate performance.

## 4.4 USING FORWARD MODELING IN MCTS

The performance metric used in Sec. 4.3 is only an intermediate metric. To finally test the performance of forward models, we plug them into a planning algorithm, e.g., Monte-Carlo Tree Search (MCTS). Note that traditional MCTS can only be applied when both the complete game state and the dynamics of the game environments are known. However, neither conditions are satisfied for RTS games, thus learned forward models and value function are necessary.

In MCTS, We use shared tree (tree parallelization) among multiple threads. This is because compared to other approaches (e.g., root parallelization Chaslot et al. (2008)), expanding a tree node, as a costly operation that requires feed-forwarding a neural network, only happens once. To increase diversity of initial exploration, we use virtual loss of size 5. During the execution of MCTS, the network performs a *projection* step that maps the input state to its latent representation, a *forwarding* step that predicts the latent state $h_{t+1}$ given $h_t$ and $a_t$, and a *prediction* step that gives the value of the predicted state for back-propagation in MCTS.

Interestingly, the models with strong performance in delayed prediction task (e.g., **MatchPi**) does not perform well in MCTS. On the contrary, **PredN** gives a consistently strong performance (25%) over 5 independently trained models, compared to the random baseline (17% win rate). Note that without a good forward model and a value function, we cannot even run reduced space MCTS. The proposed forward model makes it possible. However, compared to the performance of model-free

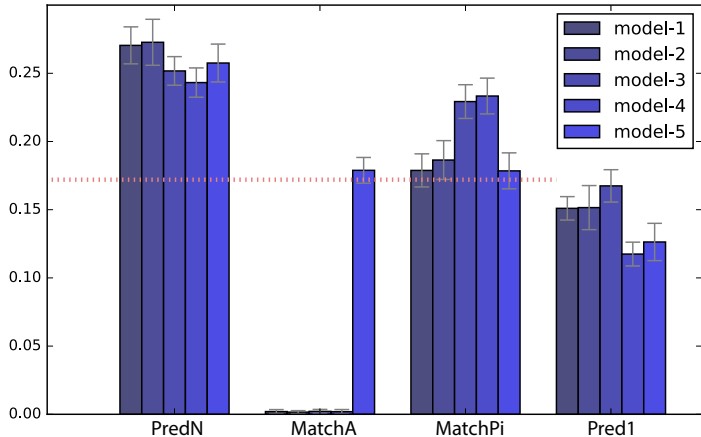

Figure 8: Win rate against `AI_SIMPLE` with frame skip 20 on 1000 games using latent space Monte-Carlo Tree Search. All results uses 100 rollouts (5 threads, each thread with 20 rollouts). The pink line is the baseline of a random agent that picks the 9 actions uniformly at each step.

approach ($> 80\%$), there is still a long way to go. We also tried removing short-term predictions (Fig. 1(f)), the performance is similar.

We have also tried combining forward model with action probability given by model-free approach using PUCT [Rosin (2011)]. However, there is no gain due to the fact that the model-free part has a much stronger performance, overshadowing the contribution of forward models.

## 5 CONCLUSION AND FUTURE WORK

Latent space learned by model-free reinforcement learning encodes important information for an agent to make sensible decisions to maximize the reward in a complicated simulated environment. In this paper, we verify the power of latent space of successfully trained model-free agent, and propose several methods to learn forward models on this space, in a real-time strategy game with incomplete information. Despite an extremely hard problem, we learn forward models that make it possible to use planning approaches such as Monte Carlo Tree Search, and show consistently positive gains over baselines.

A lot of future works follow. As a first step, although we show that it is possible to learn a forward model for incomplete information Real-time Strategy games to enable model-based planning in the latent space, it remains an open problem how to improve its performance. It is possible that despite a good forward model is learned, the value function is not good enough, e.g., putting too much focus on the on-policy trajectory, for Monte-Carlo Tree Search. Also, in this paper we use predefined 9 global actions for the game. How to automatically learn global actions from unit-based commands that are exponentially large, is still an challenging issue to solve.

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
