# OpenReview forum: "Latent forward model for Real-time Strategy game planning with incomplete information"
_ICLR.cc/2018/Conference — Reject_

### Official Review · AnonReviewer1 · 2017-11-26
**Interesting way of re-using pre-trained agents with a lot of room for improvement**

**Rating:** 5
**Confidence:** 4

**Review:**

The paper proposes to use a pretrained model-free RL agent to extract the developed state representation and further re-use it for learning forward model of the environment and planning.
The idea of re-using a pretrained agent has both pros and cons. On one hand, it can be simpler than learning a model from scratch because that would also require a decent exploration policy to sample representative trajectories from the environment. On the other hand, the usefulness of the learned representation for planning is unclear. A model-free agent can (especially if trained with certain regularization) exclude a lot of information which is potentially useful for planning, but is it necessary for reactively taking actions.
A reasonable experiment/baseline thus would be to train a model-free agent with a small reconstruction loss on top of the learned representation. In addition to that, one could fine-tune the representation during forward model training.
It would be interesting to see if this can improve the results.

I personally miss a more technical and detailed exposition of the ideas. For example, it is not described anywhere what loss is used for learning the model. MCTS is not described and a reader has to follow references and infer how exactly is it used in this particular application which makes the paper not self-contained.
Again, due to lack of equations, I don’t completely understand the last paragraph of 3.2, I suggest re-writing it (as well as some other parts) in a more explicit way.
I also could find the details on how figure 1 was produced. As I understand, MCTS was not used in this experiment. If so, how would one play with just a forward model?

It is a bit disappointing that authors seem to consider only deterministic models which clearly have very limited applicability. Is mini-RTS a deterministic environment?
Would it be possible to include a non-deterministic baseline in the experimental comparison?

Experimentally, the results are rather weak compared to pure model-free agents. Somewhat unsatisfying, longer-term prediction results into weaker game play. Doesn’t this support the argument about need in stochastic prediction?

To me, the paper in it’s current form is not written well and does not contain strong enough empirical results, so that I can’t recommend acceptance.

Minor comments:
* MatchA and PredictPi models are not introduced under such names
* Figure 1 that introduces them contains typos.
* Formatting of figure 8 needs to be fixed. This figure does not seem to be referred to anywhere in the text and the broken caption makes it hard to understand what is happening there.

---

### Official Review · AnonReviewer3 · 2017-11-27
**Interesting direction of research, but analysis is not complete and exposition is unclear.**

**Rating:** 4
**Confidence:** 5

**Review:**

Summary:

This paper studies learning forward models on latent representations of the environment, and use these for model-based planning (e.g. via MCTS) in partial-information real-time-strategy games. The testbed used is MiniRTS, a simulation environemnt for 1v1 RTS.

Forecasting the future suffers from buildup / propagation of prediction errors, hence the paper uses multi-step errors to stabilize learning.

The paper:
1. describes how to train strong agents that might have learned an informative latent representation of the observed state-space.
2. Evaluates how informative the latent states are via state reconstruction.
3. trains variatns of a forward model f on the hidden states of the various learned agents.
4. evaluates different f within MCTS for MiniRTS.

Pro:
- This is a neat idea and addresses the important question of how to learn accurate models of the environment from data, and how to integrate them with model-free methods.
- The experimental setting is very non-trivial and novel.

Con:
- The manuscript is unclear in many parts -- this should be greatly improved.
1. The different forward models are not explained well (what is MatchPi, MatchA, PredN?). Which forward model is trained from which model-free agent?
2. How is the forward model / value function used in MCTS? I assume it's similar to what AlphaGo does, but right now it's not clear at all how everything is put together.

- The paper devotes a lot of space (sect 4.1) on details of learning and behavior of the model-free agents X. Yet it is unclear how this informs us about the quality of the learned forward models f. It would be more informative to focus in the main text on the aspects that inform us about f, and put the training details in an appendix.

- As there are many details on how the model-free agents are trained and the system has many moving parts, it is not clear what is important and what is not wrt to the eventual winrate comparisons of the MCTS models. Right now, it is not clear to me why MatchA / PredN differ so much in Fig 8.

- The conclusion seems quite negative: the model-based methods fare *much* worse than the model-free agent. Is this because of the MCTS approach? Because f is not good? Because the latent h is not informative enough? This requires a much more thorough evaluation.

Overall:
I think this is an interesting direction of research, but the current manuscript does provide a complete and clear analysis.

Detailed:
- What are the right prediction tasks that ensure the latent space captures enough of the forward model?
- What is the error of the raw h-predictions? Only the state-reconstruction error is shown now.
- Figure 6 / sect 4.2: which model-free agent is used? Also fig 6 misses captions.
- Figure 8: scrambled caption.
- Does scheduled sampling / Dagger (Ross et al.) improve the long-term stability in this case?

---

### Official Review · AnonReviewer2 · 2017-11-27
**Interesting approach to learning a model, but underperforms model-free methods**

**Rating:** 4
**Confidence:** 3

**Review:**

Summary: This paper proposes to use the latent representations learned by a model-free RL agent to learn a transition model for use in model-based RL (specifically MCTS). The paper introduces a strong model-free baseline (win rate ~80% in the MiniRTS environment) and shows that the latent space learned by this baseline does include relevant game information. They use the latent state representation to learn a model for planning, which performs slightly better than a random baseline (win rate ~25%).

Pros:
- Improvement of the model-free method from previous work by incorporating information about previously observed states, demonstrating the importance of memory.
- Interesting evaluation of which input features are important for the model-free algorithm, such as base HP ratio and the amount of resources available.

Cons:
- The model-based approach is disappointing compared to the model-free approach.

Quality and Clarity:

The paper in general is well-written and easy to follow and seems technically correct, though I found some of the figures and definitions confusing, specifically:

- The terms for different forward models are not defined (e.g. MatchPi, MatchA, etc.). I can infer what they mean based on Figure 1 but it would be helpful to readers to define them explicitly.
- In Figure 3b, it is not clear to me what the difference between the red and blue curves is.
- In Figure 4, it would be helpful to label which color corresponds to the agent and which to the rule-based AI.
- The caption in Figure 8 is malformatted.
- In Figure 7, the baseline of \hat{h_t}=h_{t-2} seems strange---I would find it more useful for Figure 7 to compare to the performance if the model were not used (i.e. if \hat{h_t}=h_t) to see how much performance suffers as a result of model error.

Originality:

I am unfamiliar with the MiniRTS environment, but given that it is only published in this year's NIPS (and that I couldn't find any other papers about it on Google Scholar) it seems that this is the first paper to compare model-free and model-based approaches in this domain. However, the model-free approach does not seem particularly novel in that it is just an extension of that from Tian et al. (2017) plus some additional features. The idea of learning a model based on the features from a model-free agent seems novel but lacks significance in that the results are not very compelling (see below).

Significance:

I feel the paper overstates the results in saying that the learned forward model is usable in MCTS. The implication in the abstract and introduction (at least as I interpreted it) is that the learned model would outperform a model-free method, but upon reading the rest of the paper I was disappointed to learn that in reality it drastically underperforms. The baseline used in the paper is a random baseline, which seems a bit unfair---a good baseline is usually an algorithm that is an obvious first choice, such as the model-free approach.

---

### Author Response · Authors · 2018-01-05
**We thanks reviewers for their comments.**

We thank the reviewers for their insightful comments.

Our paper points to an interesting direction that uses learned latent space from model-free approaches as the latent space for dynamics models. In the MiniRTS game, we verified that the latent space that leads to strong performance of model-free methods is both compact and contains crucial information of the game situation, which could be interesting. We agree that the analysis can be done more thoroughly and the final performance (e.g., MCTS with learned dynamics model) is not that satisfactory, compared to model-free approaches. We will continue working on it in the future.

Details:
What is MiniRTS?

MiniRTS is recently proposed as part of ELF platform [Tian et al (NIPS 2017)]. It is a miniature 2-player real-time strategy game with basic functionality (e.g., resource gathering, troop/facilities building, incomplete information (fog of war), multiple unit types, continuous motion of units, etc).

The symbols "MatchPi", “MatchA", etc, are now defined properly in the text (paper is updated).

We have fixed the broken captions of Fig. 8.

R2:
3. In Fig. 3b, red curves are the value average on won games, while blue curves are on lost games.
4. "\hat{h_t} = h_t" would be cheating since the baseline would have access to the most recent observation, which the forward modeling does not. Note that the forward model can only access information in the previous frames, say, 2 frames ago. Performance wise, "\hat{h_t} = h_t" would yield higher performance than the learned forward model.

R3:
1. We have updated the paper to explain different training paradigms (MatchPi etc).

2. How the forward (or dynamics) model is used in MCTS:
The forward model is used to predict the future states given the current state. The predicted future state is thus used as the latent representation of child nodes, and so on. This is useful when the game is imperfect information and the game dynamics is unknown (like what MiniRTS is). In comparison, systems like AlphaGo knows the complete information and perfect game dynamics. Other than this difference, the MCTS algorithm is like what AlphaGo does: in each rollout it expands a leaf node to get its value and policy distribution, and use the value to backpropagate the winrate estimation at each intermediate nodes.

3. In Figure 6, the PrevSeen agent is used.

4. We haven't tried scheduled sampling / Dagger (Ross et al.) yet. We acknowledge that this is an interesting direction to explore.

R1:
1. Fig. 1 is an illustrative fig about different ways of training forward models. Fig. 2 is the training curves for model-free agents and no MCTS is involved.

2. MiniRTS is indeed an deterministic environment. This means that if all the initial states are fixed (including random seeds), then the game simulator will give exactly the same consequence. However, in the presence of Fog of War (each player cannot see the opponent's behavior if his troops are not nearby), the environment from one player's point of view may not be deterministic. We acknowledge that modeling uncertainty could be a good direction to work on.

---

### Decision · Program_Chairs · 2018-01-29
**ICLR 2018 Conference Acceptance Decision**

**Decision:**

Reject

**Comment:**

There was certainly some interest in this paper which investigates learning latent models of the environment for model-based planning, particularly articulated by Reviewer3.  However, the bulk of reviewer remarks focused on negatives, such as:

--The model-based approach is disappointing compared to the model-free approach.
--The idea of learning a model based on the features from a model-free agent seems novel but lacks significance in that the results are not very compelling.
--I feel the paper overstates the results in saying that the learned forward model is usable in MCTS.
-- the paper in it’s current form is not written well and does not contain strong enough empirical results